# An Overview of E-Cigarette Impact on Reproductive Health

**DOI:** 10.3390/life13030827

**Published:** 2023-03-18

**Authors:** Debbie Montjean, Marie-Hélène Godin Pagé, Marie-Claire Bélanger, Moncef Benkhalifa, Pierre Miron

**Affiliations:** 1Fertilys Fertility Center, 1950 Maurice-Gauvin Street, Laval, QC H7S 1Z5, Canada; 2Centre de Recherche du Centre Hospitalier de l’Université de Montréal (CRCHUM), CHUM Research Center, 900 Saint-Denis Street, Montreal, QC H2X 0A9, Canada; 3Médecine et Biologie de la Reproduction et Laboratoire PERITOX, Université Picardie Jules Verne, CBH-CHU Amiens Picardie, 1 Rond-Point du Professeur Christian Cabrol, 80054 Amiens, France; 4Institut National de Recherche Scientifique–Centre Armand-Frappier Santé Biotechnologie, 531 Boulevard des Prairies, Laval, QC H7V 1B7, Canada

**Keywords:** e-cigarette, reproductive health, sperm, oocyte, infertility, safety

## Abstract

Electronic cigarettes (e-cigarettes) are often considered a “safe substitute” for conventional cigarette cessation. The composition of the fluid is not always clearly defined and shows a large variation within brands and manufacturers. More than 80 compounds were detected in liquids and aerosols. E-cigarettes contain nicotine, and the addition of flavorings increases the toxicity of e-cigarette vapour in a significant manner. The heat generated by the e-cigarette leads to the oxidation and decomposition of its components, eventually forming harmful constituents in the inhaled vapour. The effects of these toxicants on male and female reproduction are well established in conventional cigarette smokers. Although toxins were measured at much lower levels in e-cigarette aerosols compared to smoke from a conventional cigarette, there are concerns about their potential impact on male and female reproduction. The information available was mainly obtained from studies conducted in animal models, and investigations in humans are scarce. However, the effects observed in animal models suggest that caution should be taken when vaping and that more research needs to be conducted to identify its potential adverse effects on fertility. The prevalence of e-cigarette usage is alarming, and warnings should be made about the impact of vaping on reproductive health. This document reviews the data regarding the impact of e-cigarette use on male and female reproduction.

## 1. Introduction 

Infertility is acknowledged by the WHO as a public health issue that affects 186 million people worldwide [1,2,3,4]. Environmental and lifestyle factors, including cigarette smoking, are known to have adverse effects on gamete quality and cause reproductive disruption [5,6]. Therefore, cigarette smoking cessation is largely recommended by professionals to individuals experiencing fertility issues. Electronic cigarettes (E-cigarettes) were used by conventional cigarette smokers as an alternative and an aid in the conventional cigarette smoking cessation process [7]. However, e-cigarette utilisation (vaping) has already shown to negatively affect general health, with adverse effects on respiratory systems, autoimmune systems, and liver functions [8,9,10]. It was also demonstrated that the addition of flavorings increased the toxicity of e-cigarette vapour in a significant manner [9]. To date, studies assessing the impact of e-cigarette utilisation on human reproduction are limited. Moreover, with an increase in the prevalence of couples turning to assisted reproductive technologies (ART) to conceive, the associated effect of e-cigarette utilisation on treatment outcomes needs to be examined. This document aims to provide an overview of the state of the art about the potential impact of e-cigarettes on female and male reproductive health.

## 2. E-Cigarettes: Types, Usage

Since they were placed on the market, e-cigarettes have undergone major evolutions (Figure 1). These devices have been called several different names and were manufactured in a large range of shapes, sizes, and types. Four generations of e-cigarettes have been developed so far (Figure 1). Overall, its main components remain unchanged and consist of a cartridge that contains a fluid, an atomizer that acts as a heating element to vapourize the e-liquid into an aerosol, a sensor that is required to turn on the device, and a battery that provides the current needed to heat the atomizer [11]. The first generation of e-cigarettes were primarily designed for “one-time use” since they were not rechargeable or refillable. The evolution of the second generation brought devices with refillable e-liquid cartridges and batteries that could be replaced. The third generation was designed to be used multiple times and permitted the customization of the substances found in the e-liquid. Pod-Mods, the fourth generation of e-cigarettes, included all the features of the previous generation, and came in a wide variety of shapes, sizes, and colors.

The e-cigarette market is in constant evolution, with more than 500 brands and 8000 flavours commercialized to date [9]. These devices have become very popular amongst young adults. Indeed, 85% of adults from 18–29 years had tried vaping in 2018 and its consumption is directly correlated with advertisement exposure (https://www.statista.com/statistics/882611/vaping-and-electronic-cigarette-use-us-by-age (accessed on 20 July 2018) [12]). However, e-cigarette prevalence dropped because of recent licensure and taxation [13]. E-cigarette usage is perceived differently between age groups. On the one hand, adolescents and young adults may consider the usage of these devices fashionable. As a matter of fact, 40% of young e-cigarette users had never smoked before. Based on this data, the e-cigarette was suggested to be an entry point to conventional cigarette smoking [14]. On the other hand, in older users, the use of e-cigarettes acted as an aid for conventional cigarette smoking cessation [14]. This stemmed from the common belief that e-cigarette utilisation is less harmful than conventional cigarette smoking. Nevertheless, e-cigarettes are composed of various toxins, exposing users to substances that are not without any impact on their general health [9].

## 3. Composition of E-Cigarettes

There are as many compositions of e-cigarettes as there are models available on the market. The main components that are consistently found, in varying concentrations, in e-liquid are water, glycols, and nicotine (Figure 2). Comprehensive analyses of e-liquid and aerosol detected more than 80 compounds [10], amongst them:

### 3.1. Glycols

Glycols are the major component of e-cigarette liquid, and the most detected are propylene glycol (1,2-propandiol) and glycerol (glycerine) [9]. Glycols are generally considered nontoxic. However, the heat induced by the e-cigarette leads to their oxidation and decomposition, leading to the formation of harmful constituents in the inhaled vapour [10,15].

### 3.2. Nicotine

Depending on the e-liquid manufacturing company, one inhale from an e-cigarette device can contain 0–35 mg of nicotine [16]. In most cases, the detected levels of nicotine were not in accordance with the concentration disclosed by the manufacturer [8,9]. Notably, nicotine was also detected in “Nicotine-Free Products” [8,9]. Due to the general lack of regulation regarding e-cigarette manufacturing, there are large discrepancies in nicotine content found within e-liquid between countries [9].

### 3.3. Particles

It is known that all tested e-cigarettes generate nanoparticles (11–600 nm) [10]. After inhalation, these particles will enter the respiratory system and can be deposited within the lungs as far as the alveolar region [10]. There are limited studies on the long-term effects of these nanoparticles. Most investigations measuring the size of the particles produced by these devices were not conclusive due to inconsistencies in the results caused by a variability in e-liquid composition between brands [11,17].

### 3.4. Metals

Many metals have been consistently detected in the vapour generated by e-cigarettes. These include lead, chromium, tin, silver, nickel, cadmium, and aluminum [9]. Alarmingly, mercury was sporadically described as a component of the vapour generated by these devices [9].

### 3.5. Tobacco-Specific Nitrosamines (TSNAs)

TSNAs are the result of processing tobacco or the addition of tobacco flavouring in e-cigarettes. They are found in the fluid and vapour of e-cigarettes in a large range of concentrations [18]. They are highly toxic, and their potential carcinogenic effects are a matter of debate [10,19,20,21].

### 3.6. Carbonyls

Carbonyls detected in e-cigarettes include butyraldehyde and aldehydes such as formaldehyde, acetaldehyde acetone, acrolein, and propionic aldehyde. These substances are known to be highly irritanting and toxic [9]. The main source of carbonyl is probably sucrose, and certain flavouring components have been shown to cause bronchiolitis and obstructive lung pathologies [22,23].

### 3.7. Volatile Organic Compounds (VOCs)

VOCs generated by the utilisation of e-cigarette comprise benzene, styrene, ethylbenzene, and toluene, all of which are classified as carcinogenic substances [24]. VOCs provoke irritation, headaches, liver, kidney, and central nervous system damages [9,10].

### 3.8. Hydrocarbons and Polycyclic Aromatic Hydrocarbons (PAHs)

Amongst PAHs, benzo(a)pyrene is known as a carcinogenic molecule. Although they were suspected to be present in flavours, their presence in e-cigarette fluid is a matter of debate. However, no health impact was described in relation to e-cigarette utilisation so far [9,10].

### 3.9. Phenols

Phenols were detected in refill solutions at different levels depending on the brand [9]. When inhaled, phenols highly irritate the skin, eyes, and mucous membranes. Long term exposure to phenols induces anorexia, progressive weight loss, diarrhea, vertigo, salivation, a dark coloration of the urine, and blood and liver effects [24].

## 4. E-Cigarette and Conventional Cigarette: Compositional Comparisons

As e-cigarettes are commonly acknowledged as a substitute for conventional cigarettes, their composition and safety have been compared to conventional cigarettes in numerous studies. However, the inconsistencies between the methodologies used (detection method, liquid or vapour, animal models, conditions) make it very difficult to draw definitive conclusions. Besides, the conduct of the comparative studies and interpretation of the results may have been biased by commercial conflicts of interest. However, the data generated by these investigations remains informative.

The effects of nicotine contained in e-cigarettes are theoretically the same as conventional cigarettes [9,10,25,26,27]. However, nicotine concentration in the e-liquid does not necessarily correlate with nicotine levels found in the vapour. Moreover, the levels of nicotine detected in e-cigarette fluid are lower than those observed in conventional cigarettes. Therefore, the nicotine toxicity risk was concluded to be modest for most e-cigarette brands compared to conventional cigarettes [28].

It was suggested that e-cigarette users were exposed to equal or higher levels of formaldehyde compared to tobacco smokers [9]. Moreover, e-cigarette particle concentrations and size range were found to be lower or comparable to conventional cigarettes [17,29]. Levels of metals were found to be equal or higher in e-cigarettes compared to conventional cigarettes [30,31]. It was shown elsewhere that although acetaldehyde, formaldehyde, TSNA, mercury, metals, carbonyls, and volatile organic compounds were found in almost all e-cigarettes, they were detected at lower levels than in conventional cigarette smoke [9,32,33].

## 5. Conventional Cigarette Smoking and Reproduction

Conventional cigarette usage was largely studied in the context of reproduction. Overall, conventional cigarette smoking is acknowledged to adversely affect reproductive systems at different levels, namely from the hypothalamic-pituitary-gonadal axis to gamete quality. Therefore, conventional cigarette cessation is highly recommended in individuals who are trying to conceive, all the more so in patients seeking fertility treatments. Table 1 provides an overview of the evidence regarding the impact of conventional cigarette smoking on male and female reproduction and ART outcomes.

## 6. Reprotoxicological Profile of E-Cigarette Components

Due to the numerous components and varying concentrations of substances found in e-liquid, the precise toxic effect of e-cigarette utilisation on reproduction is hard to determine. Each component on its own could have deleterious effects on one’s reproduction. Moreover, substance interaction adds complexity to conclusively determining the negative effects of vaping on one’s reproductive health. Generally, our knowledge of the toxic effects of e-cigarette components are known through studies of conventional cigarettes. However, as discussed above, e-cigarettes vary largely from conventional cigarettes. Comprehensive analyses have shown that many of these substances have a negative impact on reproduction, among them.

### 6.1. Nicotine

The reprotoxicity of nicotine is largely documented in relation to conventional cigarette smoking. Although knowledge about the exposure to nicotine in the context of e-cigarette utilisation is limited, it has gained interest over the past few years. Nicotine disrupts the hypothalamic-pituitary-gonadal axis in acute and chronic smokers [34,35]. The reproductive system is under the control of several sexual hormones like follicle-stimulating hormone, luteinizing hormone, sex hormone-binding globulin, and cortisol, whose levels were found to be altered in nicotine consumers [36,37]. Nicotine is also known to be a powerful vasoconstrictor that can impair sexual and erectile functions [52].

Nicotine was also shown to have an effect on reproduction at the gonadal level. Indeed, nicotine triggers oxidative stress in the testis, resulting in a global alteration of testis function, with a decrease in testosterone level, lower epididymal sperm number and viability, and increased levels of apoptosis in spermatogonia and spermatocytes [53,54,55,56]. Semen and serum nicotine levels showed a negative correlation with sperm concentration [43]. A significant decrease in sperm motility was also described in infertile and fertile men displaying high serum nicotine levels with a dose-specific effect on sperm motility and morphology [42,57]. Moreover, sperm function was not left unaffected by exposure to nicotine since lower sperm fertilizing capacity and viability were described in men with higher nicotine levels [57]. Lastly, the ultrastructure and motility machinery functions in spermatozoa were reported to be modified at a higher incidence in nicotine consumers [38].

The process of fertilization is also a target of nicotine since one of its crucial events, acrosomal reaction, was shown to be significantly altered by nicotine [44]. Nicotine reduces offspring numbers and induces abnormal and delayed implantation in e-cigarette exposed female mice. The impaired implantation seen in relation to nicotine consumption was linked to a decrease in endometrial thickness caused by impaired blood flow to the uterine tissue [58,59]. Similarly, there was a marked decrease in blood flow in both the maternal uterine and fetal umbilical circulation when females were exposed to nicotine during pregnancy. These females gave birth to pups showing significantly reduced body weight and length with behavioural changes [60,61]. Interestingly, in bovine studies, nicotine was shown to impair cellular division and chromosomal alignment, leading to a decrease in the quality and quantity of cultured blastocysts [62].

### 6.2. Flavouring Compounds

The effect of flavours added to e-cigarette fluid was assessed on reproductive ability and outcomes mainly in animal models. It was demonstrated that long-term daily exposure to nicotine-free flavoured e-cigarette vapour induced low testis weight, increased apoptosis in testes, increased oxidative stress, and an increase in the inhibition of the expression of main steroidogenesis enzymes [55,63].

Moreover, elevated sperm DNA fragmentation levels in mouse testes were also described after long-term exposure to e-cigarette flavours [63]. Bubble gum flavour was found to damage germ cells, while cinnamon altered germ-cell precursors in exposed mouse testes [63]. Moreover, increased teratozoospermia, mainly in the form of abnormal flagellum, was observed in rats exposed to e-cigarette flavours [54]. Experiments in zebra fish showed that cinnamaldehyde, a constituent of the bubble gum flavour adversely affected embryo development [64]. Similar fetal weight and crown-rump length were observed in rat newborns whose mothers were exposed to juice flavor [60].

A human study exposing spermatozoa to nicotine-free cinnamon and bubble gum flavoured e-cigarettes showed a decrease in sperm motility [63]. The reprotoxicity studies on e-cigarette flavouring compounds are scarce, and further investigations are needed, particularly in humans.

### 6.3. Heavy Metals

As a consequence of the heating process of e-fluid and the device components, e-cigarettes release numerous metal nanoparticles such as lead, nickel, chromium, aluminum, iron, copper, silver, zinc, tin, manganese, ceramic, and silica [10,61]. Although the impact of male exposure to these metals in the context of e-cigarette utilisation has not yet been proven, environmental exposure to these nanoparticles was shown to negatively affect human sperm concentration, sperm motility, and sperm function with a potential effect on fertility status [65,66,67,68,69,70]. At high concentrations, cadmium was reported to have detrimental effects on human, bovine, and murine oocyte maturation, fertilization, early cleavage, and blastocyst development rates [62,71]. Copper was also found to negatively impact embryo development in a dose-dependent manner [72]. The female reproductive system seems to also be a target of heavy metals. While the mechanism remains to be clarified, ovarian steroidogenesis, including estradiol, FSHR, StAR, CYP11A1, CYP19A1, HSD3β1 and SF-1 levels, were found to be disrupted in women and rats exposed to copper and nickel [73,74]. These observations were accompanied by increased apoptotic cell numbers and inflammation levels in the ovaries [73]. Lastly, parental occupational exposure to lead was suspected to increase the risk of spontaneous abortion and congenital malformations [75].

### 6.4. E-Cigarette Vapour

E-cigarette utilisation generates vapour that is a mixture of diverse components, including TSNA, acrolein, glycidol, formaldehyde, VOCs, and PAH [10]. The reprotoxicity of formaldehyde was largely studied in animal models. It showed the ability to alter testis structure, induce sperm parameter defects, and modify sexual behaviour [76,77]. Long-term exposure to formaldehyde increases oxidative stress, resulting in adverse effects on rat ovarian histology with a dramatic decrease in mature follicle number and size [78]. Although there have been few studies conducted in humans, formaldehyde is acknowledged as reprotoxic. A negative impact on semen motility was observed in men exposed to formaldehyde vapour at work [79]. Similarly, environmental exposure to VOCs and PAH adversely affects endocrine function and semen quality (sperm counts and morphology), ultimately causing reproductive issues [80,81,82]. VOCs are well known to have detrimental effects on embryo development resulting in decreased IVF success chances since they have lower implantation and pregnancy rates [83,84].

## 7. Evidence of the Impact of E-Cigarette Exposure on Reproduction

As previously mentioned, data about the potential impact of e-cigarette exposure on reproduction is limited. Studies tackling this topic were conducted mainly in animal models under experimental conditions that do not reproduce the utilisation of e-cigarettes in humans. However, the outcome of these studies remains informative. This section aims to provide an overview of the evidence of the impact of e-cigarette exposure on male and female gonads, gametes, the reproductive tract, and subsequently on reproduction. A summary of the proposed effects of e-cigarette-mediated reproductive disruption is available in Figure 3.

### 7.1. Evidence Analysis of the Impact of E-Cigarette on Male Reproduction

While studies on the effect of e-cigarettes on human male reproduction are limited, numerous groups have investigated their effect in animal models.

Exposure to e-cigarettes was reported to disturb the hypothalamo-pituitary axis, resulting in altered gonadal function and semen quality (Figure 3). Indeed, Wawryk-Gawda and collaborators showed that in male rats exposed to e-cigarette vapour had increased apoptosis in spermatogonia and spermatocyte, an alteration of the morphology and function of the seminiferous epithelium, as well as unica albuginea malformations [56]. Other studies linked e-cigarette utilisation with steroidogenesis disruption and global disorganisation of the testes, accompanied by significant desquamation of germ cells [53,54,55]. Moreover, low testicular weight and a higher apoptotic cell number in the testis was observed in the context of e-cigarette exposure [55,63]. Intraperitoneal injection of e-cigarette liquid in male rats induced toxicity and testicular inflammation, which, in turn, affected sperm production and sperm quality with lower sperm density, reduction of epididymal sperm number, and lower sperm viability [53,54,56]. When inhaled for 4 weeks by male rats, the same flavoring induced apoptosis in testes [63]. The sperm of rats exposed to e-cigarette vapour showed increased teratozoospermia (looped tail, flagellar angulation, and complete absence of flagellum) [54,55,56]. Studies showed that sperm chromatin integrity could also be affected by e-cigarette exposure. In fact, higher DNA damage was observed in both testis and sperm of exposed rats [55,63]. These findings suggest potential mutagenic effects of e-cigarettes on sperm.

Little to no studies have corroborated these findings in humans. A preliminary study, presented at the British Fertility Society Conference in 2017, investigating the effect of e-cigarette flavouring on human sperm, showed a significant decrease in motility in specimen cultured with e-liquid flavouring [63]. This study and the results obtained in animal models all suggest that vaping could have pathogenic effects on male reproduction and caution should be used when vaping and trying to conceive.

### 7.2. Evidence Analysis of the Impact of E-Cigarette on Female Reproduction

Evidence of the impact of e-cigarettes on female reproduction suggests that the female reproductive system is not left unaffected by exposure to e-cigarettes (Figure 3). Unlike sperm, there is no evidence linking the impact of e-cigarette utilisation on intrinsic oocyte quality and oocyte genome integrity. However, some data suggests that ovarian function is impaired in animal models exposed to e-cigarettes. Indeed, a decreased percentage of normal follicles was described in the ovaries of female rats exposed to e-cigarette fluid [85]. Hormone levels were also affected in these animals, where a reduction in estrogen secretion was observed [85]. Implantation and pregnancy outcomes were also affected in mice exposed to e-cigarette vapour. Microarray analysis showed an alteration in uterine receptivity transcripts in e-cigarette exposed mice. These females experienced a delay in embryo implantation, although the animals showed high progesterone levels, resulting in a decreased offspring number [86].

Interestingly, some studies suggest that e-cigarette exposure not only has a negative impact on one’s reproductive health but also on the offspring when exposed to e-cigarette components in utero. In fact, there was a trend towards lower fertility in male offspring and lower body weight and length in all offspring [86,87,88,89]. These findings suggest a hypothetical toxicity of e-cigarette exposure on an in utero developing fetus. Neonatal exposure to e-cigarette induced altered lung growth, weight gain with significant and persistent behavioral alterations [88]. This raises the question of the potential impact of e-cigarettes on non-users that are passively exposed to the vapour during pregnancy.

### 7.3. Evidence Analysis of the Impact of E-Cigarette on ART Outcomes

Very little is known about the true impact of e-cigarette utilisation on ART outcomes. Luckily, many studies have investigated the deleterious effect of conventional smoking in relation to ART outcomes, providing some basis for what could be expected from e-cigarette utilisation. As presented in Table 1, smoking has many negative effects on ART outcomes. In males, smoking has been shown to decrease spermogram quality and increase the risk of pregnancy loss, both of which would have a significant impact on ART outcomes [47]. In females, smoking was associated with a lower number of oocytes obtained during an oocyte retrieval procedure and a poorer response to ovarian stimulation [47,90]. Finally, in couples doing IVF, higher rates of miscarriage and lower chances of achieving pregnancy were observed in women who smoked conventional cigarettes [35,48,91]. While most studies concluded that there was a time-dependent and dose-dependent effect of smoking, all results suggested a negative impact of smoking on ART treatment outcomes.

While there are no prospective studies assessing the direct reproductive effects of e-cigarette use on ART outcomes, studies have suggested that the use of e-cigarettes can provide levels of nicotine and other metabolites that are similar to those produced by traditional cigarettes [9,17,29,30,31,32,33,92]. Therefore, it would be wise to assume that e-cigarettes would have similar negative effects on ART outcomes than those observed in traditional cigarette consumers, until further demonstrated otherwise. It would thus be beneficial to promote awareness of its potential negative impact on ART outcomes before commencing a treatment plan.

## 8. Conclusions

Studies tackling the effect of e-cigarette consumption on reproductive health mostly used animal models. Apart from some methodological inconsistencies, the data suggest that e-cigarette utilisation, whether it contains nicotine and/or flavours, leads to potential pathological alterations of reproductive functions. Observations on the impact of e-cigarette vapour exposure in neonates raises the issue of passive vaping, where pregnant individuals are exposed to the vapour released by the user. No clear conclusion can be drawn on the potential detrimental effect of vaping on reproduction and offspring health. Therefore, human studies are required to clarify this issue. The design of human studies is expected to be challenging because of the numerous confounding factors that need to be controlled for in order to obtain meaningful results. In fact, such studies will have to consider the diversity of e-cigarette devices and brands available on the market. Indeed, they all come with their own specifications in terms of composition and characteristics (ex: e-fluid heating temperature). The utilisation conditions will also need to be considered, for instance, the frequency of utilisation, or whether it is used indoors or outdoors. Such information may be collected by self-report but will have to be confirmed or correlated with objective biological markers. Furthermore, as mentioned earlier, the exact composition of e-cigarette fluid is not always disclaimed by manufacturers, or worse, it is incorrectly advertised. All these limitations will surely be circumvented, and we expect enlightening studies to feed the literature on this topic in a near future.

Overall, given the numerous potential adverse effects of e-cigarettes described in animal models, one can admit that vaping is far from a safe alternative to conventional cigarette smoking. To date, data about the impact of e-cigarette consumption on artificial reproductive technologies outcomes are also lacking. However, the evidence of the impact of utilisation of these devices is alarming, and people trying to conceive should be aware of their potential impact on their reproductive health. Moreover, when patients are undergoing assisted reproductive treatments, they should be warned about the potential impact of e-cigarette usage on treatment outcomes and possibly on the health of their children. 

## Figures and Tables

**Figure 1 life-13-00827-f001:**
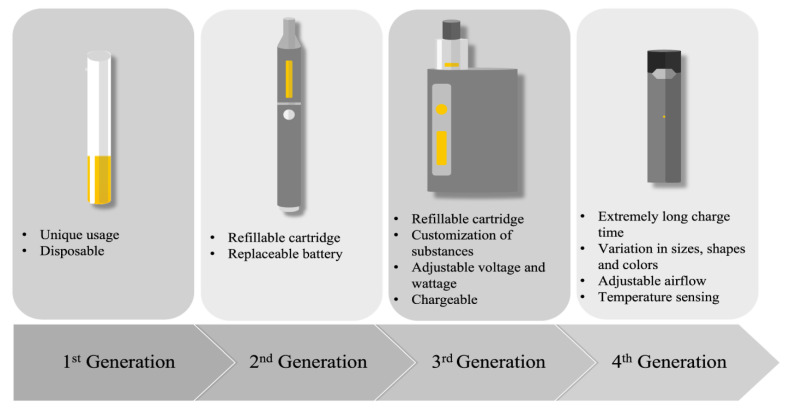
Description of the four generations of e-cigarettes.

**Figure 2 life-13-00827-f002:**
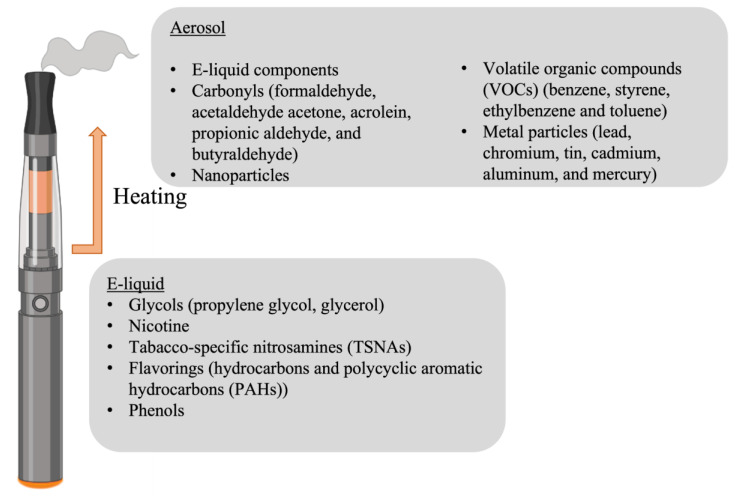
Components described in e-cigarette fluid and aerosol.

**Figure 3 life-13-00827-f003:**
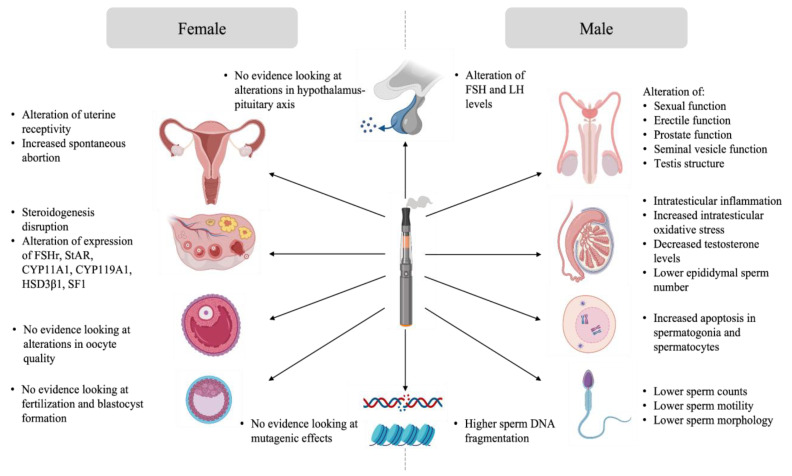
Effects of e-cigarette-mediated reproductive disruption.

**Table 1 life-13-00827-t001:** Evidence analysis of the impact of conventional cigarettes on male and female reproduction, and ART outcomes.

Outcome	Evidence	References
Conventional cigarette and male reproduction	Disruption Hypothalamic-pituitary-gonadal axis	[34,35]
	Alteration of Follicle-stimulating hormone, luteinizing hormone, sex hormone-binding globulin and cortisol levels Alteration of testicular endocrine function Alteration of secretory function of Sertoli and Leydig cells	[35,36,37,38]
	Alteration of structure and function of epididymis	[38]
	Alteration of prostate and seminal vesicle function Modification of seminal fluid composition Alteration of sperm parameters	[39]
	Increased risk of erectile dysfunction	[40,41]
	Reduction of sperm concentration Reduction sperm motility Increased teratozoospermia Reduction of sperm vitality Reduction of fertilizing capacity Reduction in sperm acrosome reaction	[37,38,42,43,44]
	Higher oxidative stress in seminal fluid Higher sperm DNA fragmentation level Higher incidence of point mutations and epimutation in sperm	[37,38,45,46]
	No impact on time to pregnancy No impact on birth rate	[45]
	Higher incidence spontaneous abortion Higher risk for progeny to present facial anomalies, neural tube defects, spina bifida, anencephaly	[37]
Conventional cigarette smoking and female reproduction	Reduction of ovarian reserve Higher FSH level	[47,48]
	Lower progesterone in luteal phase Increased level of androgens Higher level of LH at menses Increased risk of anovulation Transgenerational adverse effect on fertility status and semen parameters	[35,49]
Conventional cigarette smoking and ART outcomes	Negative impact on oocyte morphology	[50]
	Female/Male smoking: No impact on fertilization rate, embryo quality No impact on clinical pregnancy and live birth chances	[35,51]
	Female smoking: Lower number of recovered oocytes Poor response to stimulation Lower fertilization rate Fewer embryos Lower chances to achieve pregnancy	[48]
	Increased risk of miscarriage Increased risk of total fertilization failure	[35]
	Male smoking caused an increased risk of pregnancy loss	[47]

## Data Availability

Not applicable.

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
