# Peer review of "An Overview of E-Cigarette Impact on Reproductive Health"

_life, 2023, doi:10.3390/life13030827_

Round 1

Reviewer 1 Report

The authors purport to evaluate the effect of e-cigarettes on reproduction and ART results.  The lack of human data substantially limit any discussion of the effects of ecigarettes on reproduction or ART.  So, the title (and to a large degree the conclusions) are not relevant to the data presented.

Rewrite of the title and emphasis of the article is recommended.

For each statement, the level of evidence should be provided.

Mechanisms of potential mechanisms in rodent studies should be presented separately from the limited human data

Data on ART effects should be explicitly outline including effects on live births, clinical pregnancy, embryo development & implantation, sperm DNA fragmentation and fertilization results.

Although the presented data are interesting, they do not rise to the level of a meta-analysis of existing information.  There is a great deal of non-reproductive toxicity discussed as well.  The alarms for extensive use of these materials should be tempered by the limited data presented.

Author Response

Reviewer 1:

  1. Comment:

The authors purport to evaluate the effect of e-cigarettes on reproduction and ART results.  The lack of human data substantially limit any discussion of the effects of ecigarettes on reproduction or ART.  So, the title (and to a large degree the conclusions) are not relevant to the data presented.

Response:

There are limited studies on the effect of e-cigarette on reproduction in humans. However, some animal studies were performed and the results were discussed in this review. The title of this review does not mention the overview of the effect of e-cigarette on reproduction in humans specifically.

  1. Comment:

Rewrite of the title and emphasis of the article is recommended.

Response:

The reviewers concern about the relevance of the title was considered and the title of the review was changed to “An overview of e-cigarette impact on reproductive health” to better reflect what is discussed in the article.

  1. Comment:

For each statement, the level of evidence should be provided.

Response:

Due to the lack of specificity of this statement, no major modification was made in the review to address this comment. However, a revision of all the statements made to ensure adequate reference to findings in the literature was performed and some references were added.

  1. Comment:

Mechanisms of potential mechanisms in rodent studies should be presented separately from the limited human data.

Response:

We suspect that this comment is relating to section 7 “Evidence of the impact of e-cigarette exposure on reproduction” of the review, where findings in animal models and humans are discussed together. No significant body of literature is available to allow for a separate section on human data. However, a separate paragraph was introduced in section 7.a “Evidence analysis of the impact of e-cigarette on male reproduction” recapitulating that results in animal models provide probable cause to be cautious of vaping when trying to conceive. Moreover, a preliminary study, concluding the effect of e-cigarette favoring on sperm motility in humans, previously discussed with findings in animal models, has been added to this new paragraph to separate findings in animal model to those in humans.

  1. Comment:

Data on ART effects should be explicitly outline including effects on live births, clinical pregnancy, embryo development & implantation, sperm DNA fragmentation and fertilization results.

Response:

Similarly to the comment above, there are no studies identifying the direct effect of e-cigarette utilisation on ART outcomes. A clear discussion addressing this point was added under section 7.c “Evidence analysis of the impact of e-cigarette on ART outcomes”.

  1. Comment:

Although the presented data are interesting, they do not rise to the level of a meta-analysis of existing information.  There is a great deal of non-reproductive toxicity discussed as well.  The alarms for extensive use of these materials should be tempered by the limited data presented.

Response:

This article is a review article of the present data available in field of reproduction and ART. Some non-reproductive toxicity was added to emphasize its global negative effect on human health and to compare its usage to conventional cigarettes. The mention of limited studies providing a clear and conclusive stance was raised at numerous occasions and all caution was given as recommendations and not as conclusions.  

Reviewer 2 Report

I am very grateful for the invitation to review the manuscript. The topic of this article is very interesting, the logical structure and language of the article is clear and easy to understand, but there are still some issues that need to be clarified.

1. The quality of the evidence for each cited article needs to be marked in Table 1, so that it provides more useful information for the reader.

2. The authors should elaborate more on the effects of e-cigarette components on sperm-oocyte binding, embryo development and embryo implantation, the current content is evidently enough.

3. Authors stated that all images were created with BioRender.com. Therefore, the certificate from the BioRender.com should be provided.

Author Response

Reviewer 2

  1. Comment:

The quality of the evidence for each cited article needs to be marked in Table 1, so that it provides more useful information for the reader.

Response:

This is a very relevant comment. This exercise would be uninformative and unconclusive since it is essentially experimental studies in animals that do not necessarily reproduce the real conditions of use of the e-cigarettes.

  1. Comment:

The authors should elaborate more on the effects of e-cigarette components on sperm-oocyte binding, embryo development and embryo implantation, the current content is evidently enough.

Response:

  • The effect on nicotine consumption on sperm acrosomal reaction was already discussed. The addition of nicotine effect on embryo implantation and the expansion of its effect on embryo development has been added to section 6.a.
  • The effects of metal elements and VOCs on oocyte quality and embryo development is now addressed in sections 6.c and 6.d

  1. Comment:

Authors stated that all images were created with BioRender.com. Therefore, the certificate from the BioRender.com should be provided.

Response:

The certificate will be provided with revision submission.

Reviewer 3 Report

The manuscript entitled "An overview of e-cigarette impact on reproduction and ART outcomes" has been reviewed, and the following results were attained:

The manuscript first introduces various types of e-cigarettes, their composition, their toxic effects, possible considerations, a comparison with conventional smoking, and finally their effects on reproduction. The manuscript's language is fluid and involving and has provided reasonable insight into the basics of e-cigarettes and possible usage outcomes.

However, the following opinions could be implemented to improve the manuscript:

1. As the title states, it is recommended to explain the assisted reproductive treatment concept in a separate paragraph. Also, assisted reproductive technology should be introduced first in the text.

2. The provided illustrations are well-designed, but the quality is not adequate.

3. The bullet points in Table 1 need alignment for better readability.

4. The spelling of "rat" in line 265 needs revision.

5. Authors' contributions, funding statement, conflicts of interest statement, data availability statement, informed consent statement, and abbreviation section should be added as well according to the publisher's requirements.

Author Response

Reviewer 3

  1. Comment :

As the title states, it is recommended to explain the assisted reproductive treatment concept in a separate paragraph. Also, assisted reproductive technology should be introduced first in the text.

Response:

As mentioned for Reviewer 1, a clear discussion addressing assisted reproductive treatments was added under section 7.c “Evidence analysis of the impact of e-cigarette on ART outcomes”.

Moreover, a sentence in the introduction was added to introduce assisted reproductive technologies.

  1. Comment:

The provided illustrations are well-designed, but the quality is not adequate.

Response:

PDF conversion was performed again to ensure adequate image quality.

  1. Comment:

The bullet points in Table 1 need alignment for better readability.

Response:

The modification was made.

  1. Comment:

The spelling of "rat" in line 265 needs revision.

Response:

The correction was made.

  1. Comment:

Authors' contributions, funding statement, conflicts of interest statement, data availability statement, informed consent statement, and abbreviation section should be added as well according to the publisher's requirements.

Response:

All sections mentioned were added to the article after the conclusion section.

Round 2

Reviewer 1 Report

No changes made in the abstract.

Author Response

The manuscript has been corrected consequently

Reviewer 3 Report

The corrections are sufficient for acceptance.

Author Response

The manuscript has been correctement consequently